# Improvement and Performance Evaluation When Implementing a Distance-Based Registration

**Jang Hyun Baek**

Department of Industrial & Information Systems Engineering and the RCIT, Jeonbuk National University, Jeonju 54896, Korea; jbaek@jbnu.ac.kr; Tel.: +82-63-270-2330

**Abstract:** An efficient location registration scheme is essential to continuously accommodate the increasing number of mobile subscribers and to offer a variety of multimedia services with good quality. The objective of this study was to analyze the optimal size for the location area of a distance-based registration (DBR) scheme by varying the number of location areas on a cell-by-cell basis, not on a ring-by-ring basis. Using our proposed cell-by-cell distance-based registration scheme with a random walk mobility model, a variety of circumstances were analyzed to obtain the optimal number of cells for location area for minimizing the total signaling cost on radio channels. Analysis results showed that the optimal number of cells for location area was between 4 and 7 in most cases. Our cell-by-cell distance-based location registration scheme had less signaling cost than an optimal ring-by-ring distance-based location registration scheme with an optimal distance threshold of 2 (the optimal number of cells for location area was 7). Therefore, when DBR is adopted, it must be implemented with an LA increasing on a cell-by-cell basis to achieve optimal performance.

**Keywords:** location registration; distance-based registration; cell-by-cell location area; semi-Markov process; optimal location area





## 1. Introduction

To cope with the continuous increase of mobile subscribers and provide various multimedia services with good quality, the efficiency of radio channels, which are limited resources, must be maximized. Due to the nature of the mobile network, two functions must be performed to connect calls to user equipment (UE): location registration, performed by the UE, and paging, performed by the base station (BS). Location registration refers to a process to update the location area (LA) information of the subscriber in the system database when the LA of the subscriber changes in the mobile network. Paging refers to a process in which the network pages the UE to identify the exact location (BS) of the UE in order to connect the incoming call to the UE. Because future mobile networks deal with smaller cells, higher user density, and higher mobility, it is essential to adopt effective mobility management schemes for location registration and paging [1–5].

Many studies have been performed to reduce the signaling cost on radio channels in terms of location registration and paging. Studies on location registration have been most frequently performed regarding zone-based registration (ZBR) [2,3,5–8] because ZBR shows a rather good performance and its implementation in real networks is easy. In addition, studies on distance-based registration (DBR) [9–15], movement-based registration (MBR) [9,16–19], time-based registration [20], tracking area list (TAL)-based registration [4,21–23], and auxiliary location registrations such as power-on registration, power-off registration, and implicit location registration have also been performed for next-generation networks [1,5]. Studies on the problem of paging load optimization [24,25] can also be included in the studies on location registration by default since they basically relate to the trade-off between paging load and registration load closely.

To analyze the performance of location registration, the mobility model of subscribers is very important. Thus, the objective of this study was to analyze the performance

of DBR based on a 2-D random walk model. Previous studies on DBR using random walk mobility models [9,11,15,16] have presented the optimal size of the LA assuming that an LA is composed of rings of cells to compare the performance with ZBR [6] or MBR [9,16]. However, this kind of ring-by-ring system environment assumed in previous studies [6,12,16,17] has significant problems for real-world systems since the number of cells comprising the LA increases discretely and steeply.

In this study, we considered DBR, which is known to have better performance than ZBR, assuming a random walk mobility model and an LA increasing on a ring-by-ring basis [6]. In this study, we proposed a DBR with an LA increasing on a cell-by-cell basis, rather than a conventional LA increasing on a ring-by-ring basis, and obtained the optimal number of cells of the LA that minimizes signaling cost on the radio channels. Throughout this study, we will show that the optimal LA can be constructed in an environment that is not possible if only a ring-by-ring DBR is considered and therefore, for an accurate evaluation of DBR, the performance should be evaluated for the LA increasing on a cell-by-cell basis.

This paper is organized as follows: Section 2 describes DBR; Section 3 describes the system environment and mobility model for analyzing location registration and paging cost, as well as the LA configuration according to the increase of the number of cells in the LA; Section 4 presents an analytical model for obtaining signaling cost on radio channels; Section 6 presents numerical results; Section 7 describes conclusions and discussion.

## 2. System Environments and Distance-Based Registration

### 2.1. System Environments

A single location area (LA) in a mobile network typically consists of multiple cells. In this study, we assumed that mobile networks were composed of cells of the same shape. In order to ideally express the range of radio waves from the base station, the shape of the cell must be close to a circle. A hexagonal cell was adopted considering the overlap of neighboring circular cells. Figure 1 illustrates a cell with a hexagonal shape.

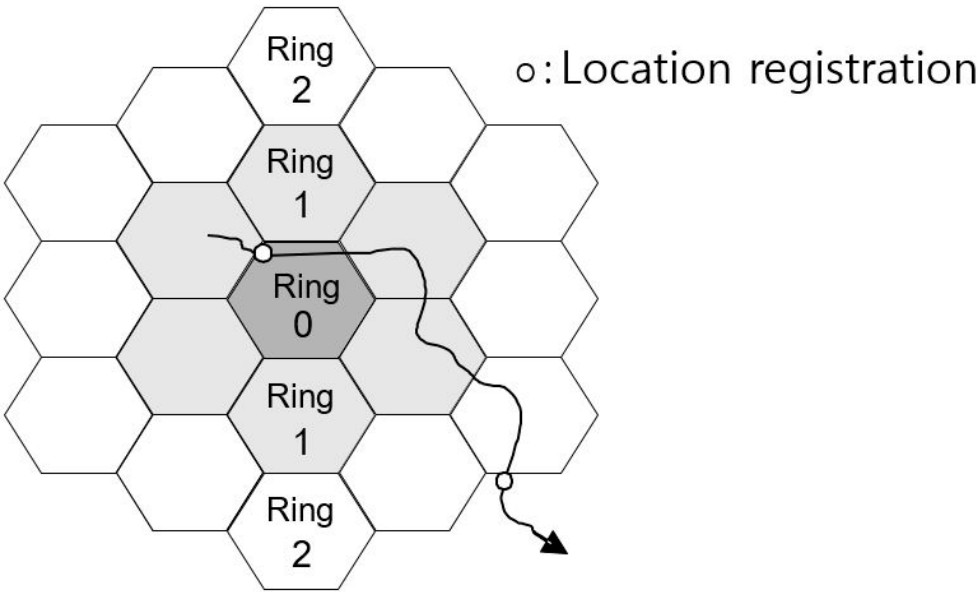

**Figure 1.** Location area of distance-based registration (D = 3).

### 2.2. Mobility Model

To analyze the performance of location registration, the mobility model of subscribers is very important. In this study, we analyzed the performance of a distance-based location registration based on a 2-D random walk model used in many studies [9,11,15,16].

The 2-D random walk mobility model assumes that the probability of choosing one of six surrounding cells is the same for all surrounding cells. For example, if a UE in ring 0 in

Figure 1 moves to a neighboring cell, it will move to one of the six cells in ring 1. In this case, the probability of the UE to move to one cell out of six surrounding cells is equally 1 out of 6 for all six cells.

### 2.3. Distance-Based Registration

If DBR is adopted, whenever the UE enters a cell, it calculates the distance between the center (X, Y) of the newly entered cell and the center (*Xc*, *Yc*) of the current LA and registers its location if the value is greater than distance threshold *D*. In other words, location registration is performed when the following expression is satisfied:

$$\sqrt{(X - X_c)^2 + (Y - Y_c)^2} \geq D \tag{1}$$

If DBR is adopted, unlike zone-based registration, there is no ping-pong phenomenon that crosses boundaries between LAs to cause multiple location registrations. In addition, unlike zone-based registration, where location registration loads are concentrated in the boundary cell of the LA, location registration loads occur equally for all cells in the network. Furthermore, if DBR is adopted, the efficiency can be improved by setting different distance thresholds for each UE using characteristics of movement or call occurrence of each UE.

Above all, when a call occurs, the current location of the UE is updated in the system through call processing messages, resulting in the same effect as the location registration in the current cell. On the other hand, a zone-based registration does not have the effect of an implicit location registration because location information does not change even if a call occurs. Therefore, in environments where calls are usually frequent, a DBR may show a better performance than a zone-based location registration [11,15,16].

### 2.4. Problem of Distance-Based Registration and Its Improvement

Previous studies [11,15,16] used distance threshold *D*, which is defined as the number of cells between the central cell of the LA and the nearest outer cell in which the UE should register its location, as shown in Figure 1.

In this definition, the distance between neighboring cells is 1. If *D* = 3, then the LA consists of a central cell (ring 0), with six cells surrounding the central cell (ring 1) and 12 cells surrounding the cells belonging to ring 1 (ring 2). When the UE moves to the outer cell of the current LA to perform location registration, a new LA is set where the cell the UE registered becomes the new central cell of the new LA.

In a hexagonal cell environment, assuming that an LA consists of *D* rings (ring 0, ring 1, ring 2, . . . , ring *D* − 1), *g*(*i*), the number of cells in the *i*-th ring, can be expressed as follows:

$$g(i) = \begin{cases} 1, & i = 0 \\ 6i, & i = 1,\ 2,\ 3,\ \ldots,\ D-1 \end{cases} \tag{2}$$

If the distance threshold is *D*, the total number of cells in the LA can be expressed as follows:

$$S = 1 + \sum_{i=1}^{D-1} 6i = 1 + 3D(D-1) \tag{3}$$

Assuming such a system environment, the number of cells in an LA is 1 if *D* is 1. It will be 7 if *D* is 2 and 19 if *D* is 3. If we assume a system environment in the form of a ring, an LA cannot be defined in which the number of cells is 2–6, 8–18, 20–36, and so on.

Previous studies [11,15,16] have shown that the optimal *D* with minimal signaling cost is 2 in most cases. However, those studies did not consider all possible LAs in terms of the number of cells comprising the LA. Thus, the optimal *D* might not be 2. To more accurately identify the optimal environment for DBR, the system environment should be assumed to include the LA between *D* = 1 and *D* = 2, between *D* = 2 and *D* = 3, and so on. In other words, performance should be analyzed when the number of cells comprising the location area is 2–6, 8–18, and so on.

In this study, we proposed a method to define the LA of DBR when the number of cells in an LA was increased one by one and determined the optimal number of cells that

made up the LA through performance analysis. Through this process, we showed that an LA increasing on a cell-by-cell basis had a smaller signaling cost than an LA increasing on a ring-by-ring basis that previous studies considered. Thus, an LA should increase on a cell-by-cell basis in order to achieve optimal performance.

### 3. DBR with an LA Increasing on a Cell-by-Cell Basis

In this study, instead of defining the system on a ring-by-ring basis, $D$, we analyzed DBR by defining it as a continuously increasing number of cells $N$. In other words, previous ring-based DBR has defined the center of the central cell of the LA as the center of the LA. Thus, the number of cells comprising the LA is 1, 7 (=1 + 6), 19 (=1 + 6 + 12), and so on. In this case, $N = 2\sim6$ (between $D = 1$ and 2), $N = 8\sim18$ (between $D = 2$ and 3), and so on cannot be determined.

In this study, to consider all cases, we did not simply limit the center of the LA to the center of the cell. Instead, we tried to have an LA consisting of a variety of cells by newly defining the center of the LA to be possible at any point in the LA. This study dealt with cases where $N = 1\sim12$ for convenience of analysis. Note that previous studies [11,15,16] have shown that the optimal $D$ with minimal signaling cost is mostly 2. In this study, on the other hand, we found that the signaling cost appeared to be smaller for $N = 1\sim12$ compared to $D = 2$.

As one example for the new definition of the center of the LA, let us look at the case when the number of cells, $N$, is 4 in DBR. To explain the proposed method, let us denote two cells close to the center of the LA (C), as 0 and the remaining two cells as 1 for convenience as shown in Figure 2 below.

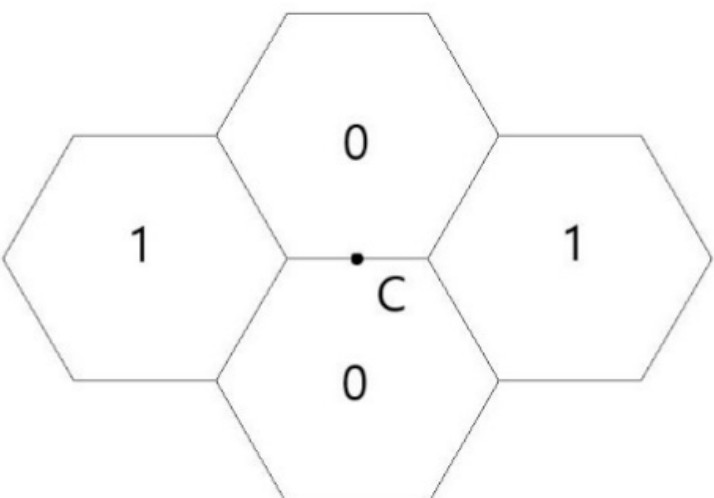

**Figure 2.** Center of location area ($N = 4$).

In a ring-by-ring DBR, when entering a new cell, the distance between the center of the entered cell and the center of the central cell of the LA is calculated and the location registration is performed if the calculated distance is greater than $D$. However, if $N = 4$, location registration cannot be defined by such a datum. Therefore, if $N = 4$, the UE calculates the distance between the center of the entered cell and the newly defined center of the LA (C). If $N = 4$, the center of LA, C, is defined as the center of the two cells marked with 0. If the calculated distance is greater than the reference distance $D_4$, then location registration is performed.

What value should the reference distance $D_4$ have in this case? The reference distance $D_4$ should have a value between the lower bound ($L_4$) and the upper bound ($U_4$). As shown in Figure 3, $L_4$ is the distance between the center of the current LA (C) and the center

of the farthest cell from C. In addition, $U_4$ is the distance between the center of the current LA (C) and the center of the nearest cell (from C) that is not included in the current LA.

$$L_4 < D_4 < U_4$$

when $N = 4$, assuming one side is 1 in a hexagonal cell, then $L_4 = \frac{3}{2}$ and $U_4 = \frac{\sqrt{21}}{2}$ can be obtained.

$$\frac{3}{2} < D_4 < \frac{\sqrt{21}}{2} = 2.29$$

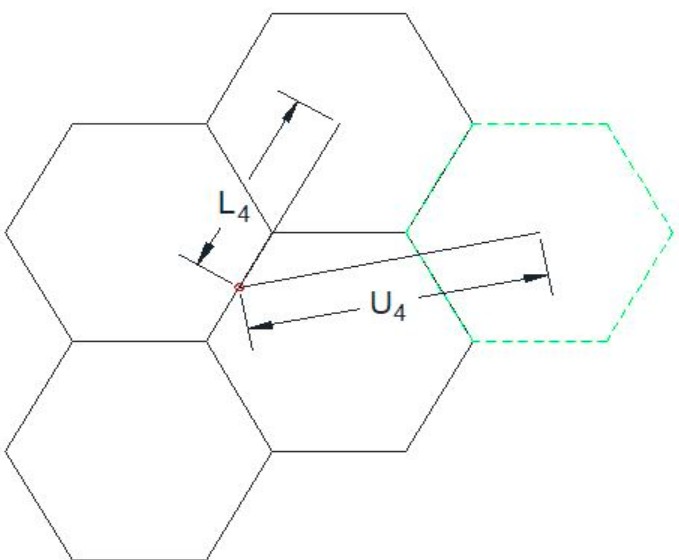

**Figure 3.** Lower bound and upper bound of reference distance ($N = 4$).

For example, if we set $D_4 = 2$, a DBR for $N = 4$ can be defined.

As one more example, we can look at a more complex case of $N = 8$. To explain the proposed method, let us denote one cell containing the center of the LA (C) as 0 for convenience, the farthest cells from C as 4, and six cells surrounding 0 as 1, 2, and 3 by two cells each as shown in Figure 4 below.

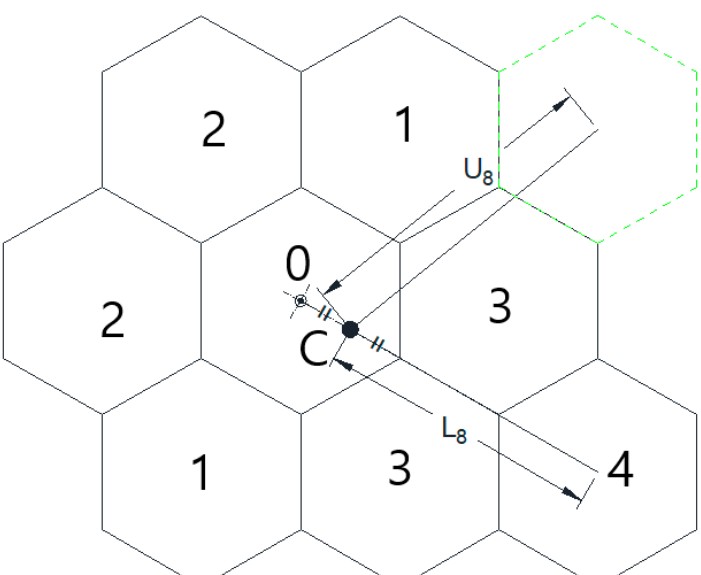

**Figure 4.** Center of location area, lower bound, and upper bound of reference distance ($N = 8$).

In a ring-by-ring DBR, if $N = 8$, location registration cannot be defined. If $N = 8$, the UE calculates the distance between the center of the entered cell and the newly defined center of the LA (C). If $N = 8$, the center of the LA, C, is defined as the midpoint between the center of the cell 0 and the vertex (of cell 0) directed for the farthest cell (4). If the calculated distance is greater than the reference distance $D_8$, then the location registration is performed.

If $N = 8$, the reference distance $D_8$ should have the value between the lower bound ($L_8$) and the upper bound ($U_8$). $L_8$ is the distance between the center of current LA (C) and the center of the farthest cell from C. $U_8$ is the distance between the center of the current LA (C) and the center of the nearest cell (from C) that is not included in the current LA.

$$L_8 < D_8 < U_8$$

when $N = 8$, assuming one side is 1 in a hexagonal cell, then $L_8 = \frac{5}{2}$ and $U_8 = \frac{\sqrt{31}}{2}$ can be obtained.

$$\frac{5}{2} < D_8 < \frac{\sqrt{31}}{2} = 2.78$$

For example, if we set to $D_8 = 2.6$, a DBR for $N = 8$ can be defined.

Figure 5 shows the LA and cell 0 (central cell) of the LA for a cell-by-cell DBR for $N = 4$ and $N = 8$. In Figure 5a, when a UE exits the current LA, it registers its new LA such that the cell in which it registers is one of the central cells of the new LA as shown in the figure. Similarly, in the case of $N = 8$, a UE registers its new LA such that the cell in which it registers is the central cell of the new LA as shown in Figure 5b.

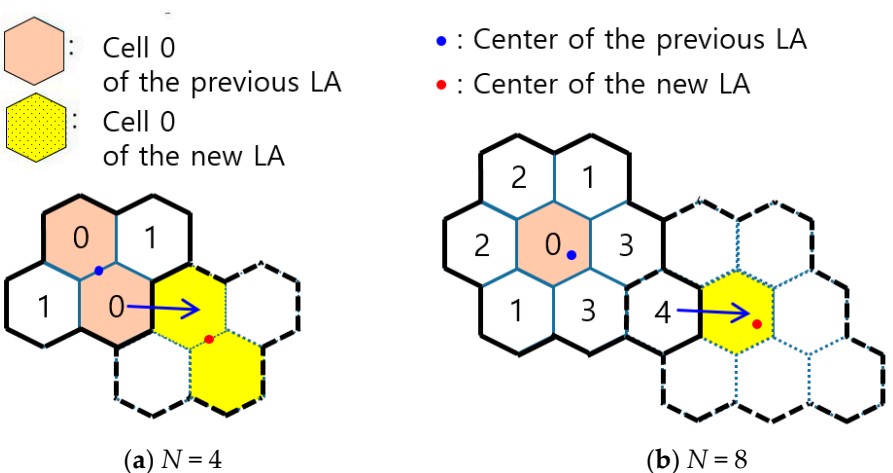

(**a**) $N = 4$  (**b**) $N = 8$

**Figure 5.** Registration and new location area in a cell-by-cell distance-based registration.

Similarly, we can define the configuration of an LA and the center of the LA for various $N$ to calculate the range of each reference distance.

Table 1 below summarizes the configuration of the LA with increasing $N$, the center of the LA, and the range of the reference distance ($L_N < D_N < U_N$).

Newly defining the center of the LA as shown above enables the formation of an LA increasing on a cell-by-cell basis, making it possible to mitigate a sudden increase in signaling cost of the ring-by-ring DBR.

**Table 1.** Configuration of the LA and range of the reference distance for various *N*.

| N | Configuration of LA | Range of Reference Distance |
|:---:|:---:|:---:|
| 2 |  | $\frac{\sqrt{3}}{2} < D_2 < 1.5$ |
| 3 |  | $2 < D_3 < 1$ |
| 4 |  | $\frac{3}{2} < D_4 < \frac{\sqrt{21}}{2}$ |
| 5 |  | $\frac{\sqrt{13}}{2} < D_5 < \frac{\sqrt{19}}{2}$ |



**Table 1.** *Cont.*

| N | Configuration of LA | Range of Reference Distance |
|---|---|---|
| 6 |  | $\frac{\sqrt{63}}{4} < D_6 < \frac{5\sqrt{3}}{4}$ |
| 7 |  | $\sqrt{3} < D_7 < 3$ |
| 8 |  | $\frac{\sqrt{31}}{2} < D_8 < 2.5$ |

**Table 1.** *Cont.*

| N | Configuration of LA | Range of Reference Distance |
|---|---|---|
| 9 | | $\frac{\sqrt{111}}{4} < D_9 < \frac{\sqrt{147}}{4}$ |
| 10 | | $\frac{3\sqrt{3}}{2} < D_{10} < \frac{\sqrt{39}}{2}$ |
| 12 | | $\sqrt{7} < D_{12} < \sqrt{13}$ |

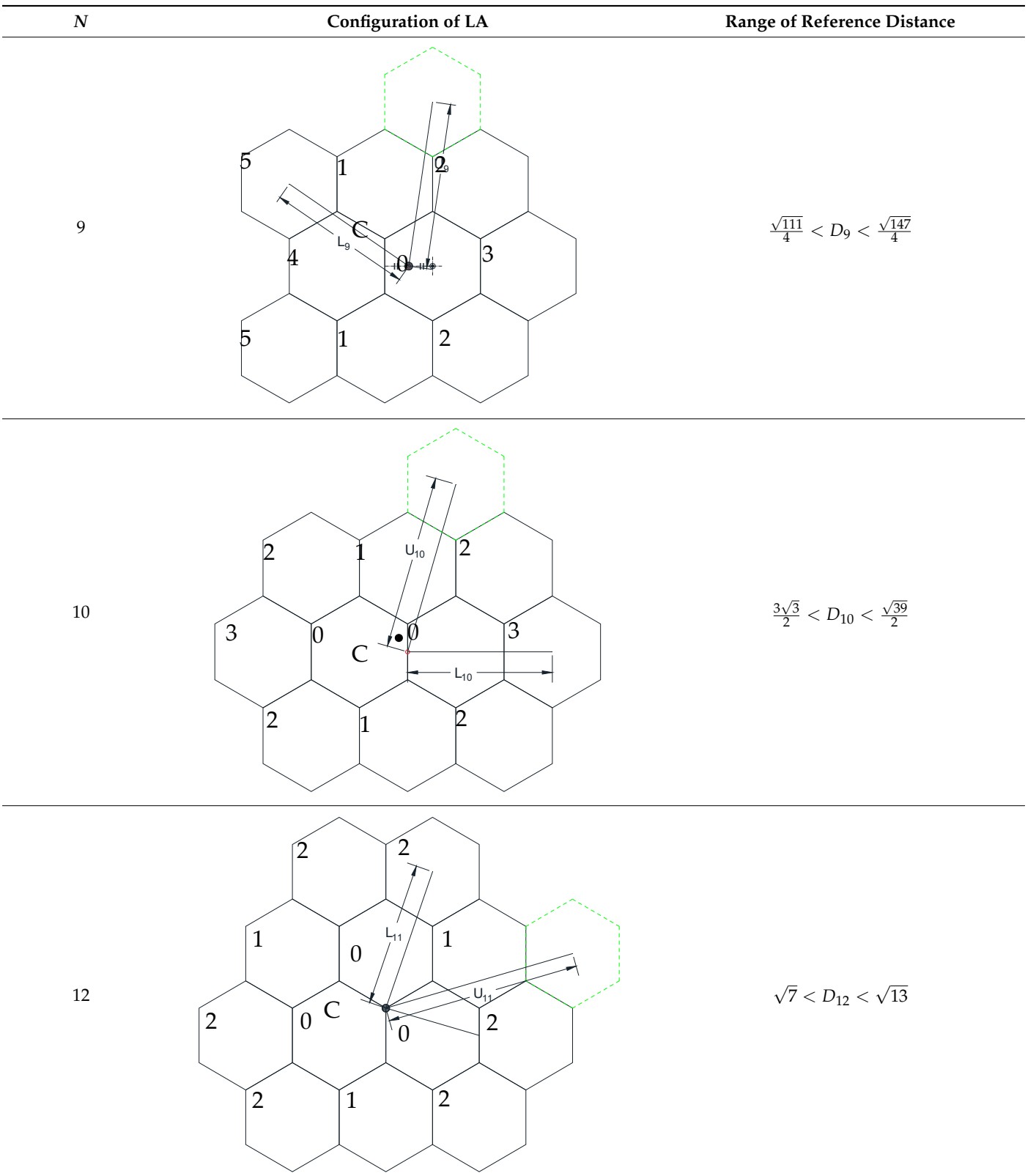

## 4. Performance Analysis

In this section, a semi-Markov process model is presented to obtain the total signaling cost for DBR when an LA is organized according to the method presented in Section 3.

The following notations are defined to analyze the total cost on radio channels:

$C_U$: Location registration cost in one hour

$C_V$: Paging cost in one hour

$U$: The location registration cost for one registration

$P$: The paging cost for one cell

$N$: Number of cells in an LA

$T_c$: Interval between two calls (r. v., $T_c \sim Exp[1/\lambda_c]$, $E(T_c) = 1/\lambda_c$)

$T_m$: Staying time in a cell (r. v., $E(T_m) = 1/\lambda_m$)

$R_m$: Time between the arrival of the call, and the time when the UE moves out of the cell

$f_m^*(s)$ The Laplace-Stieltjes Transform for $T_m$ $(= \int_{t=0}^{\infty} e^{-st} f_m(t) dt)$

We also assume the following to obtain the total cost on radio channels:

- When the UE enters a neighboring cell, the probability of selecting one of the neighboring cells is 1 out of 6.
- The incoming and outgoing calls are generated with rates $\lambda_i$ and $\lambda_o$, respectively, according to the Poisson processes and the staying time in a cell, while $T_m$ follows a general distribution with the mean $1/\lambda_m$.

Note that by the additional property of the Poisson processes, the incoming calls with the rate $\lambda_i$ and outgoing calls with the rate $\lambda_o$ will form total calls with a rate $\lambda_c$ (= $\lambda_i$ + $\lambda_o$) [26].

## 5. Registration Cost

Assuming the registration cost for one registration is $U$, then registration cost in an hour, $C_U$, is as follows:

$$C_U = U \sum_{b \in B} \widetilde{\pi_b} \cdot p_{b0} \cdot \lambda_m \tag{4}$$

$B$: set of boundary cells;

$\widetilde{\pi_b}$: steady-state probability of a boundary cell $b$;

$p_{b0}$: transition probability from boundary cell b to central cell 0.

To obtain the registration cost using this equation, the key is to obtain the steady state probability of each cell. For example, if $N = 5$, the steady state probability of each cell in the LA can be obtained. If $N = 5$, three states can be defined as shown in Table 1 and Figure 6.

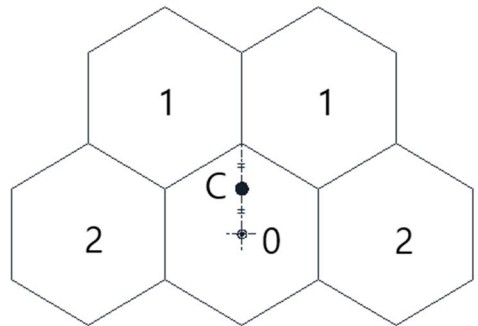

**Figure 6.** States of each cell in the location area for $N$ = 5.

State 0: The state when the UE is in the central cell is marked as '0';

State 1: The state when the UE is in boundary cells is marked as '1';

State 2: The state when the UE is in boundary cells is marked as '2';

State 0′: The state when a call occurs to or from the UE, the cell is changed to the central cell.

UE in state 0 can move to state 1 with a probability of $(1/3)P[T_c > T_m]$. It can move to state 2 with a probability of $(1/3)P[T_c > T_m]$, or move to a new LA (and be in state 0 again) with a probability of $(1/3)P[T_c > T_m]$. When a call occurs, the UE in state 0 can transit to state 0′ with a probability of $P[T_c < T_m]$. Note that state 0′ is related to implicit registration.

Similarly, the UE in state 1 can enter a state 0 with a probability of $(1/6)P[T_c > T_m]$. It can move to state 2 with a probability of $(1/6)P[T_c > T_m]$, or move to a new LA (and be in state 0 again) with a probability of $(1/2)P[T_c > T_m]$. When a call is generated, the UE in state 1 can also transit to state 0′ with a probability of $P[T_c < T_m]$.

The UE in state 0′ can enter state 1 with a probability of $(1/3)P[T_c > R_m]$. It can move to state 2 with a probability of $(1/3)P[T_c > R_m]$, or move to a new LA (and be in state 0) with a probability of $(1/3)P[T_c > R_m]$. Finally, when a call is generated, the UE in state 0′ can transit to state 0′ again with a probability of $P[T_c < R_m]$.

For convenience, using the following characters:

$$m = P[T_c > T_m], \quad m' = P[T_c > R_m] \tag{5}$$

we can get the final state transition diagram as shown in Figure 7. The corresponding transition probability matrix is shown in Figure 8.

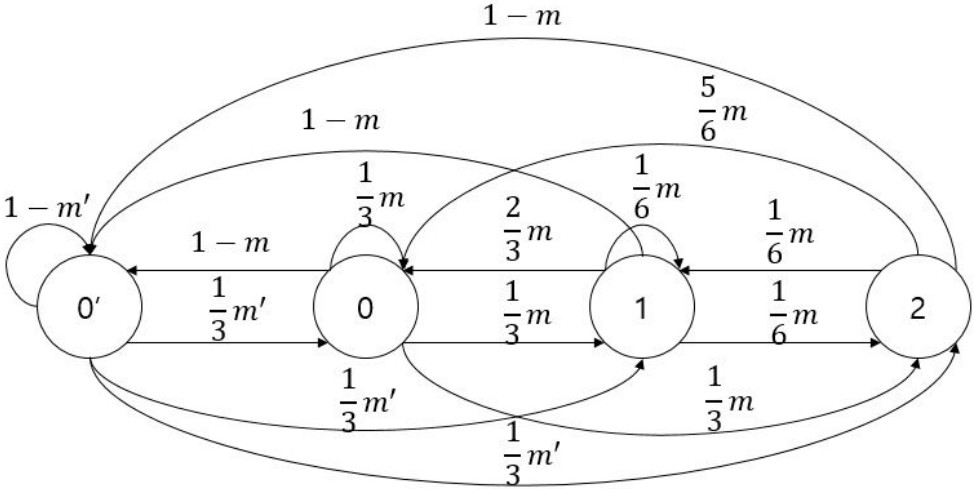

**Figure 7.** State transition diagran ($N$ = 5).

$$P = \begin{bmatrix} \frac{1}{3}m & \frac{1}{3}m & \frac{1}{3}m & 1-m \\ \frac{2}{3}m & \frac{1}{6}m & \frac{1}{6}m & 1-m \\ \frac{5}{6}m & \frac{1}{6}m & 0 & 1-m \\ \frac{1}{3}m' & \frac{1}{3}m' & \frac{1}{3}m' & 1-m' \end{bmatrix}$$

**Figure 8.** Transition probability matrix.

Similarly, we can get the state transition diagram and the corresponding transition probability matrix for various $N$ to calculate the steady-state probability of each state.

Table 2 below summarizes the state of each cell in an LA and the corresponding transition probability matrix for various $N$.

**Table 2.** State of each cell in location area and transition probability matrix for various *N*.

| *N* | Definition of States | Transition Probability Matrix P |
|---|---|---|
| 2 |  | $\begin{bmatrix} m & 1-m \\ m' & 1-m' \end{bmatrix}$ |
| 3 |  | $\begin{bmatrix} m & 1-m \\ m' & 1-m' \end{bmatrix}$ |
| 4 |  | $\begin{bmatrix} \frac{2}{3}m & \frac{1}{3}m & 1-m \\ m & 0 & 1-m \\ \frac{2}{3}m' & \frac{1}{3}m' & 1-m' \end{bmatrix}$ |
| 5 |  | $\begin{bmatrix} \frac{1}{3}m & \frac{1}{3}m & \frac{1}{3}m & 1-m \\ \frac{2}{3}m & \frac{1}{6}m & \frac{1}{6}m & 1-m \\ \frac{5}{6}m & \frac{1}{6}m & 0 & 1-m \\ \frac{1}{3}m' & \frac{1}{3}m' & \frac{1}{3}m' & 1-m' \end{bmatrix}$ |
| 6 |  | $\begin{bmatrix} \frac{1}{6}m & \frac{1}{3}m & \frac{1}{6}m & \frac{1}{3}m & 1-m \\ \frac{2}{3}m & 0 & \frac{1}{6}m & \frac{1}{6}m & 1-m \\ \frac{2}{3}m & \frac{1}{3}m & 0 & 0 & 1-m \\ \frac{5}{6}m & \frac{1}{6}m & 0 & 0 & 1-m \\ \frac{1}{6}m' & \frac{1}{3}m' & \frac{1}{6}m' & \frac{1}{3}m' & 1-m' \end{bmatrix}$ |

**Table 2.** *Cont.*

| N | Definition of States | Transition Probability Matrix P |
|---|---|---|
| 7 |  | $\begin{bmatrix} 0 & m & 1-m \\ \frac{2}{3}m & \frac{1}{3}m & 1-m \\ 0 & m' & 1-m' \end{bmatrix}$ |
| 8 |  | $\begin{bmatrix} 0 & \frac{1}{3}m & \frac{1}{3}m & \frac{1}{3}m & 0 & 1-m \\ \frac{2}{3}m & 0 & \frac{1}{6}m & \frac{1}{6}m & 0 & 1-m \\ \frac{2}{3}m & \frac{1}{6}m & \frac{1}{6}m & 0 & 0 & 1-m \\ \frac{1}{2}m & \frac{1}{6}m & 0 & \frac{1}{6}m & \frac{1}{6}m & 1-m \\ \frac{2}{3}m & 0 & 0 & \frac{1}{3}m & 0 & 1-m \\ 0 & \frac{1}{3}m' & \frac{1}{3}m' & \frac{1}{3}m' & 0 & 1-m' \end{bmatrix}$ |
| 9 |  | $\begin{bmatrix} 0 & \frac{1}{3}m & \frac{1}{6}m & \frac{1}{3}m & \frac{1}{6}m & 0 & 1-m \\ \frac{2}{3}m & 0 & \frac{1}{6}m & \frac{1}{6}m & 0 & 0 & 1-m \\ \frac{2}{3}m & \frac{1}{3}m & 0 & 0 & 0 & 0 & 1-m \\ \frac{1}{2}m & \frac{1}{6}m & 0 & 0 & \frac{1}{6}m & \frac{1}{6}m & 1-m \\ \frac{1}{3}m & 0 & 0 & \frac{1}{3}m & 0 & \frac{1}{3}m & 1-m \\ \frac{2}{3}m & 0 & 0 & \frac{1}{6}m & \frac{1}{6}m & 0 & 1-m \\ 0 & \frac{1}{3}m' & \frac{1}{6}m' & \frac{1}{3}m' & \frac{1}{6}m' & 0 & 1-m' \end{bmatrix}$ |

**Table 2.** *Cont.*

| N | Definition of States | Transition Probability Matrix P |
|---|---|---|
| 10 | (Hexagonal cell diagram) Top row: 2, 1, 2; middle row: 3, 0, C, 0, 3; bottom row: 2, 1, 2 | $$\begin{bmatrix} \frac{1}{6}m & \frac{1}{3}m & \frac{1}{3}m & \frac{1}{6}m & 1-m \\ \frac{2}{3}m & 0 & \frac{1}{3}m & 0 & 1-m \\ \frac{2}{3}m & \frac{1}{6}m & 0 & \frac{1}{6}m & 1-m \\ \frac{2}{3}m & 0 & \frac{1}{3}m & 0 & 1-m \\ \frac{1}{6}m' & \frac{1}{3}m' & \frac{1}{3}m' & \frac{1}{6}m' & 1-m' \end{bmatrix}$$ |
| 12 | (Hexagonal cell diagram) Top row: 2, 2; next: 1, 0, 1; next: 2, 0, C, 0, 2; bottom: 2, 1, 2 | $$\begin{bmatrix} \frac{1}{3}m & \frac{1}{3}m & \frac{1}{3}m & 1-m \\ \frac{2}{3}m & 0 & \frac{1}{3}m & 1-m \\ \frac{2}{3}m & \frac{1}{6}m & \frac{1}{6}m & 1-m \\ \frac{1}{3}m' & \frac{1}{3}m' & \frac{1}{3}m' & 1-m' \end{bmatrix}$$ |

Now, let us obtain m and m′ as well as the steady-state probability. The probability that a UE enters other cells before a call occurs can be calculated as below [2,3]:

$$m = P[T_c > T_m] = \int_0^\infty \int_{t_m}^\infty \lambda_c e^{-\lambda_c t_c} f_m(t_m) dt_c dt_m = f_m^*(\lambda_c) \tag{6}$$

To calculate $m' = P[T_c > R_m]$, let us consider the density function of $R_m$ first. The density function of $Rm$, $f_r(t)$ is from a random observer property [26],

$$f_r(t) = \lambda_m \int_{\tau=t}^\infty f_m(\tau) d\tau = \lambda_m (1 - F_m(t)) \tag{7}$$

The Laplace–Stieltjes transform for the distribution is shown as follows:

$$\begin{aligned} f_r^*(s) &= \int_{t=0}^\infty e^{-st} f_r(t) dt = \int_{t=0}^\infty e^{-st} \lambda_m (1 - F_m(t)) dt = \frac{\lambda_m}{s} - \int_{t=0}^\infty e^{-st} \lambda_m F_m(t) dt \\ &= \frac{\lambda_m}{s} + \left[\frac{\lambda_m}{s} e^{-st} \lambda_m F_m(t)\right]_{t=0}^\infty - \frac{\lambda_m}{s} \int_{t=0}^\infty e^{-st} f_m(t) dt = \frac{\lambda_m}{s}(1 - f_m^*(s)) \end{aligned} \tag{8}$$

This gives:

$$m' = P[T_c > R_m] = \int_0^\infty \int_{t_c=r_m}^\infty \lambda_c e^{-\lambda_c t_c} f_r(r_m) dt_c dr_m = \int_0^\infty f_r(r_m) e^{-\lambda_c t_c} dr_m = \frac{\lambda_m}{\lambda_c}(1 - f_m^*(\lambda_c)) \tag{9}$$

In this case, the staying time in state $0'$ is different from the staying time in other states. The staying time in state $0'$ is the interval from the time a call to/from a UE occurs in a cell until the time it transits (moves to a neighboring cell or generates a call again). On the other hand, the staying time in other states, except for state $0'$, is the interval from the time the UE enters a cell until the time it transits (moves to a neighboring cell or generates a call).

To accurately evaluate the performance, the fact that the staying time in each state is different should be considered. First, the staying time in state $0'$ can be expressed as follows:

$$\begin{array}{ll} T_c, & if \ T_c \leq R_m \\ R_m, & if \ T_c > R_m \end{array}$$

Its mean can be obtained as follows:

$$\begin{aligned} \tau_{0'} &= \int_0^\infty \int_0^{r_m} t_c f_c(t_c) dt_c f_r(r_m) dr_m + \int_0^\infty \int_{r_m}^\infty r_m f_c(t_c) dt_c f_r(r_m) dr_m \\ &= \int_0^\infty [\int_0^{r_m} t_c f_c(t_c) dt_c + \int_{r_m}^\infty r_m f_c(t_c) dt_c] f_r(r_m) dr_m = \frac{1}{\lambda_c}(1 - \frac{\lambda_m}{\lambda_c}[1 - f_m^*(\lambda_c)]) \end{aligned} \tag{10}$$

Next, the staying time in other states, except for state $0'$, can be expressed as follows:

$$\begin{array}{ll} T_c, & if \ T_c \leq T_m \\ T_m, & if \ T_c > T_m \end{array}$$

Its mean $\tau_i$ ($i = 0, 1, 2, \dots$ ) can be derived as follows:

$$\begin{aligned} \tau_i &= \int_0^\infty \int_0^{t_m} t_c f_c(t_c) dt_c f_m(t_m) dt_m + \int_0^\infty \int_{t_m}^\infty t_m f_c(t_c) dt_c f_m(t_m) dt_m \\ &= \int_0^\infty [\int_0^{t_m} t_c f_c(t_c) dt_c + \int_{t_m}^\infty t_m f_c(t_c) dt_c] f_m(t_m) = \frac{1}{\lambda_c}(1 - f_m^*(\lambda_c)) \end{aligned} \tag{11}$$

In order to obtain the steady-state probability $\widetilde{\pi}$, the steady-state probability $\pi$ for the usual Markov chain with transition probability $P$ should be calculated first using balanced equations [26] as follows:

$$\pi P = \pi, \qquad \sum_j \pi_j = 1 \tag{12}$$

Considering different staying time, the final steady-state probability of the semi-Markov process can be obtained as follows [26]:

$$\widetilde{\pi}_i = \frac{\pi_i \tau_i}{\sum_j \pi_j \tau_j} \tag{13}$$

*5.1. Paging Cost and Total Cost*

When paging all cells in an LA simultaneously, the paging cost $C_v$ can be obtained simply as follows:

$$C_V = V \cdot N \tag{14}$$

where $V$ is the paging cost per cell and $N$ is the number of cells that make up the LA.

The final total signaling cost is the sum of the location registration cost and the paging cost:

$$C_T = C_U + C_V = U \sum_{b \in B} \widetilde{\pi_b} \cdot p_{b0} \cdot \lambda_m + V \cdot N \tag{15}$$

where $\lambda_m$ is the number of entering cells in one hour.

## 6. Numerical Results

This section analyzes registration and paging costs in DBR using the mathematical model presented in Section 4. The environment used in the analysis is assumed to be the same as in previous studies [6,9,10,15,16]:

- UE's sojourn time in a cell follows an exponential distribution with mean $1/\lambda_m$
- Arrival process of incoming calls to UE follows a Poisson process with rate $\lambda_c$
- Registration cost for one registration, $U$, is 4
- Paging cost for one cell, $V$, is 1

As a basis for comparison, call-to-mobility ratio (CMR) is used. It represents the mobility of UEs and arrival characteristics of calls. CMR stands for $\lambda_c/\lambda_m$. A CMR of 1/3 means that the UE crosses the cell boundary an average of three times between call arrivals. The smaller the CMR, the greater the mobility of the UE between call arrivals, and the more frequent the location registration occurs.

Figure 9 shows registration cost, paging cost, and total signaling cost when CMR is 1 according to a change in the number of cells in the LA. In general, the larger the number of cells in an LA, the smaller the UE's location registration and the smaller the location registration cost.

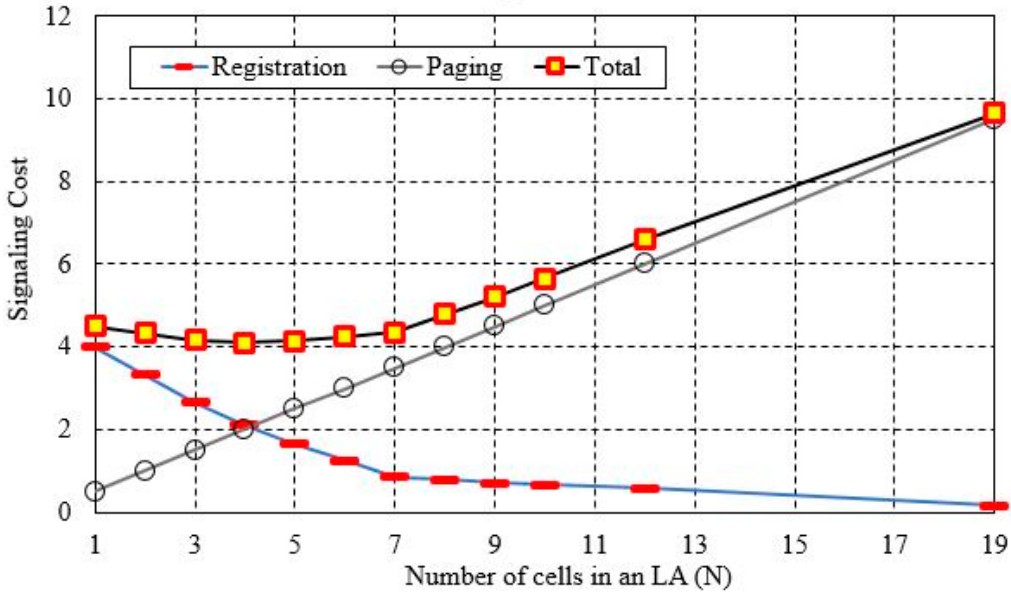

**Figure 9.** Total signaling cost for various numbers of cells in an LA (CMR = 1).

On the other hand, the larger the number of cells in an LA, the higher the number of cells that need to be paged and the higher the paging cost. Therefore, the optimal *N* that the total signaling cost is minimal can be obtained using the trade-off between the registration cost and the paging cost.

In the same analytical environment, the optimal LA size was *D* = 2 (*N* = 7) in previous studies [11,15,16]. However, in this study, assuming an LA is increasing on a cell-by-cell basis, the number of cells with a minimum total signaling cost appears to be 4~7. This means that the optimal LA can be constructed in an environment that is not possible if only a ring-by-ring DBR is considered.

Therefore, for an accurate evaluation of DBR, the performance should be evaluated for the LA increasing on a cell-by-cell basis as shown in this study.

Figure 10 shows the total signaling cost according to changes in CMR. If the CMR is 0.4, the total signaling cost represents the minimum value when *N* is 7. If the CMR is 1, the total signaling cost represents the minimum value when *N* is 4. This is because when CMR increases, paging will occur frequently, resulting in a larger role of paging cost in the total signaling cost and a smaller optimal *N*.

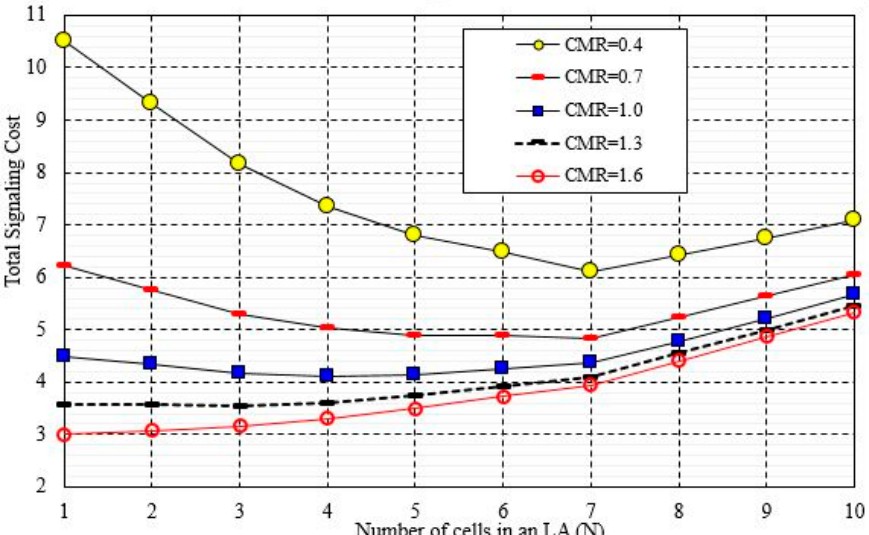

**Figure 10.** Total signaling cost for various CMRs.

Figure 11 shows total signaling cost when the registration cost U changes. When U is 6 or 8, the total signaling cost is minimal when N is 7. When U becomes smaller, the optimal N with the minimal total cost also becomes smaller. As shown in the figure, when U is 4, the total signaling cost is minimal when N is 4. When U is 2, the total signaling cost is minimal when N is 1. In other words, the smaller the registration cost U, the smaller the LA with the minimal total cost. As the registration cost U decreases, the smaller the proportion of the registration cost. Conversely, the larger the paging cost, the smaller the number of cells in the LA with the minimal cost. In general, the smaller the proportion of registration cost, the smaller the number of cells in the LA. Similarly, the greater the proportion of registration cost, the larger the optimal number of cells in the LA and the larger the total cost.

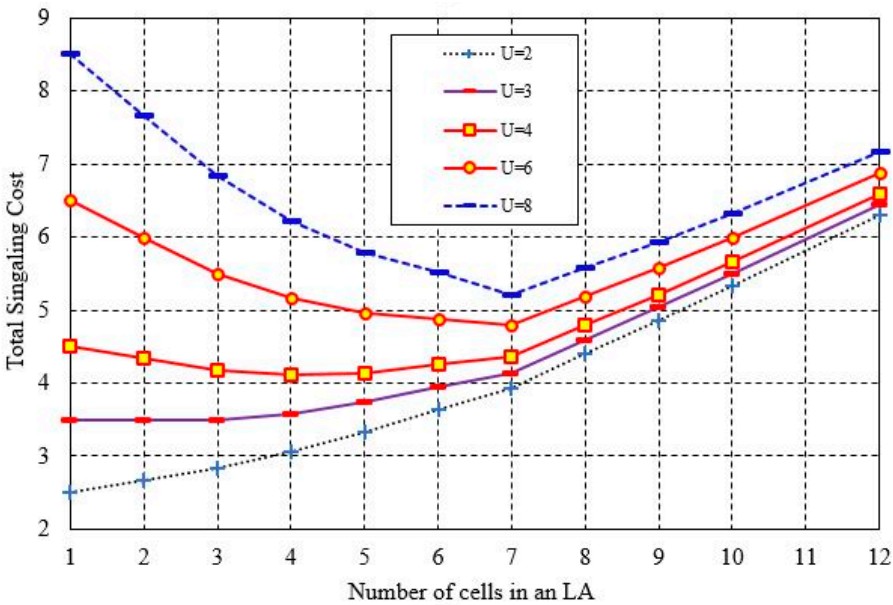

**Figure 11.** Total signaling cost for various Us (CMR = 1/3).

Figure 12 shows the total signaling cost when variance (Var) of cell sojourn time varies. As shown in the figure, even if the cell sojourn time is the same, the total signaling cost decreases as the variance increases. As the variance increases, the number of registrations decreases while the optimal N increases which minimizes the total signaling cost.

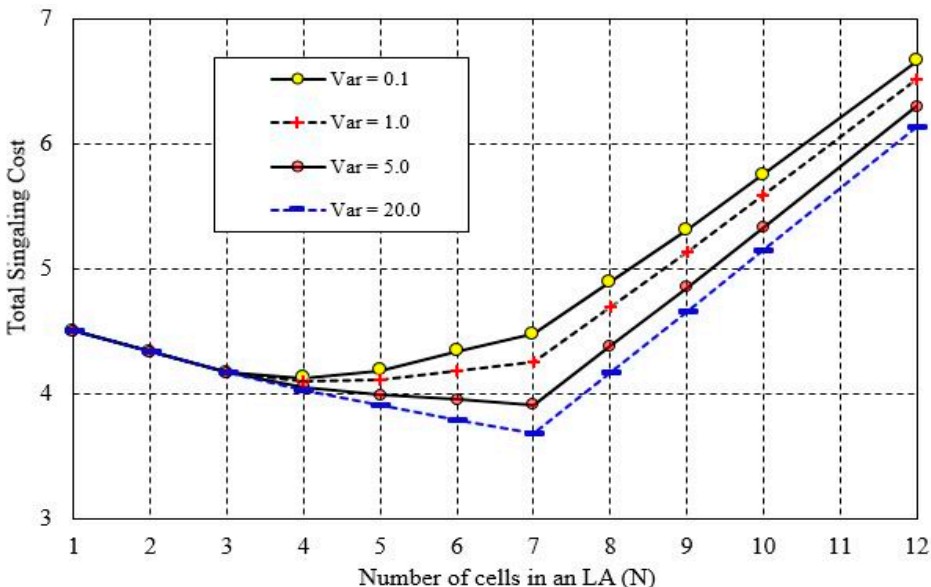

**Figure 12.** Total signaling cost for various numbers of cells in an LA (CMR = 1).

Figure 13 shows the total signaling cost when cell sojourn time E(Tm) changes. When E(Tm) is 0.6, the total signaling cost is minimal when *N* is 7. When E(Tm) increases, *N* with minimal total signaling cost is gradually reduced. When E(Tm) is 1.0, the total signaling cost is minimal when *N* is 4. When E(Tm) is 1.2, the total signaling cost is minimal when *N* is 3. In other words, as E(Tm) increases, the size of the LA with the minimum total signaling cost becomes smaller and the proportion of the total signaling cost decreases. Conversely, as E(Tm) increases, the proportion of the paging cost increases and the number of cells in the LA with the minimum cost becomes smaller.

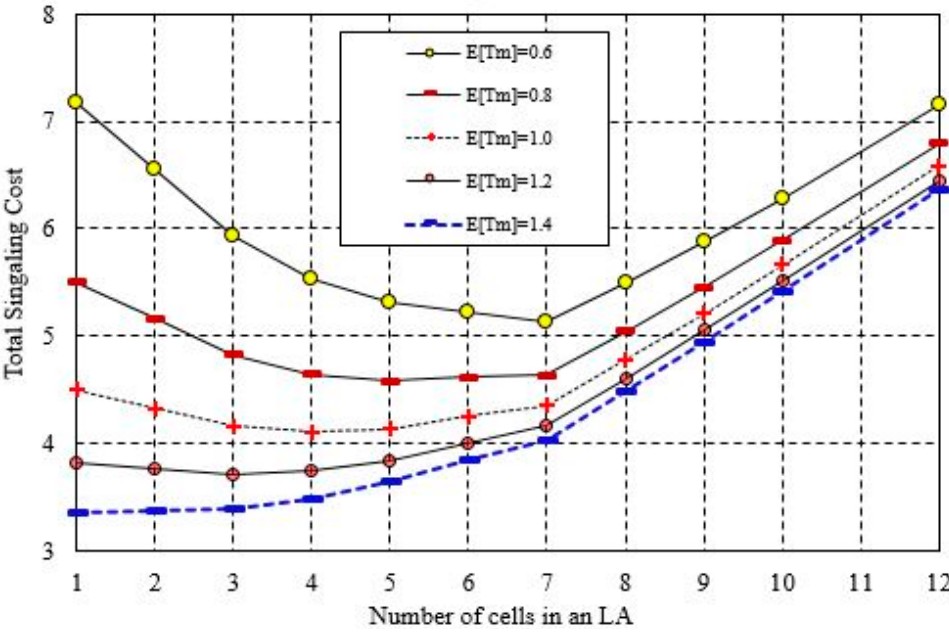

**Figure 13.** Total signaling cost for various E(T$_m$) (CMR = 1/3).

Figure 14 shows the effect of an implicit registration on reduction of signaling cost. If an implicit registration is not adopted, the total signaling cost is minimal when *N* is 4. However, if implicit registration is adopted, the total signaling cost is minimal when *N* is 7. In other words, implicit registration can reduce the location registration cost, which

increases the size of the LA and minimizes the total signaling cost. The effect of such implicit registration is greater when the call occurs frequently (i.e., the CMR becomes larger).

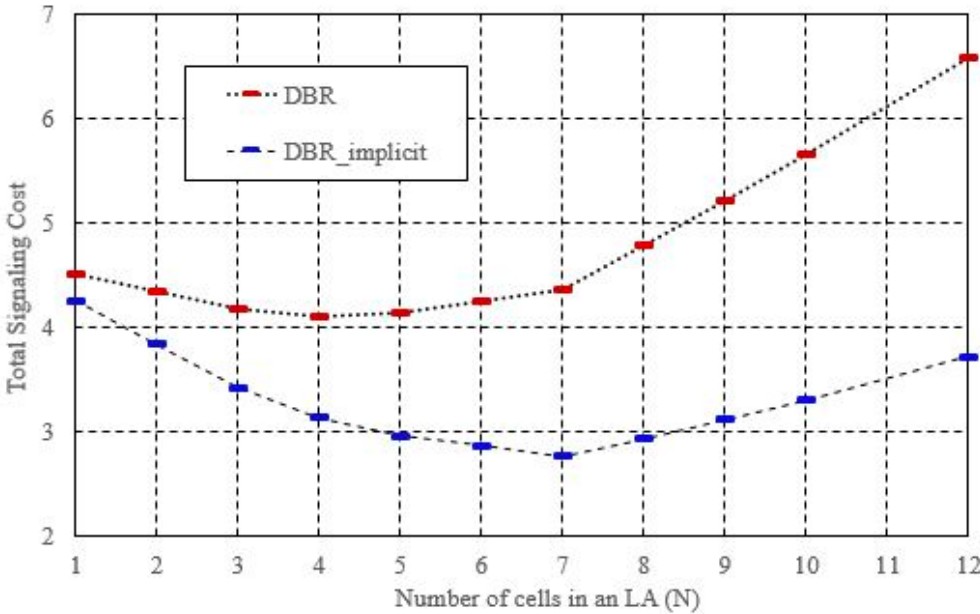

**Figure 14.** Total signaling cost for various numbers of cells in an LA (CMR = 1).

Figure 15 shows the performance comparison results of DBR and ZBR. In the previous study [6], only DBR with an LA increasing on a ring-by-ring basis was considered, so performance was compared by assuming that the LA of ZBR, the target of comparison, also increasing on a ring-by-ring basis. According to the results of previous study [6], it was analyzed that the performance of DBR was superior to that of ZBR in terms of total signaling cost when the distance threshold $D = 2$ at which total signaling cost of DBR was minimized.

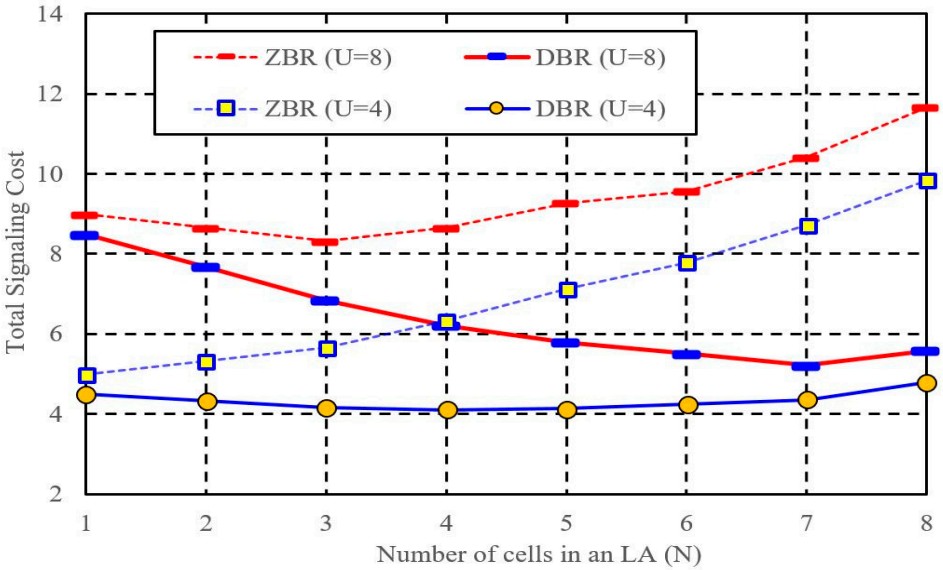

**Figure 15.** Performance comparison results of DBR and ZBR (CMR = 1).

In Figure 15, considering the DBR having an LA that increases on a cell-by-cell basis, and assuming that the LA of ZBR, the target of comparison, also increases on a cell-by-cell basis, the performance is compared. From the figure, when U = 8, DBR is minimized at $N = 7$ ($D = 2$) like the previous study [6] and shows better performance than ZBR.

However, in the previous study [6], even though the ZBR was minimized at $N = 3$, the performance of DBR was compared with the result for $D = 2$ ($N = 7$) in ZBR and concluded to be significantly better. On the other hand, when U = 4, DBR is minimal at $N = 4$ and still shows better performance than ZBR. As such, the results of accurately comparing the performance by reflecting the LA increasing on a cell-by-cell basis are new results that cannot be obtained in previous study.

In the figure, the performance of DBR is always better than that of ZBR because a pure random walk mobility model [9,11,15,16] is assumed. However, if a mobility model favorable to ZBR is assumed as in [6,7], the results may change.

Let us briefly consider the mobility model proposed in [6,7]. Figure 16 represents a revised random walk mobility model of a UE for each $N$. In the figure, $q$ is the probability that a UE can move to surrounding cells. If a pure random walk mobility model is assumed, $q$ is 1/6. However, since the boundary of the LA in ZBR is generally composed of areas such as roads, bridges, rivers and mountains where there is relatively little traffic, $q$ can be less than 1/6.

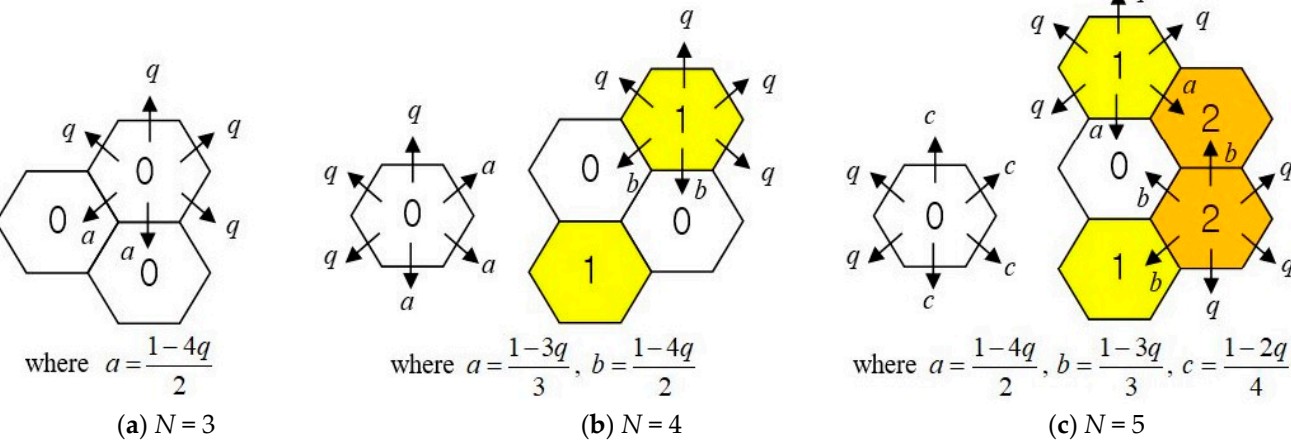

**Figure 16.** Revised random walk mobility model for ZBR [7].

Figure 17 shows the total signaling cost of ZBR for various $q$ to compare with that of DBR. In the figure, when $q = 1/12$, the performance of ZBR is better than that of DBR since the minimum of ZBR at $N = 2$ is less than that of DBR at $N = 4$. As shown in Figures 15 and 17, since the performance of the location registration method varies depending on the mobility model, network environment and so on, 4G and 5G networks adopt TAL-based location registration, which can be operated as DBR as well as ZBR [4,21–23].

In summary, the performance of location registration methods is significantly affected by the mobility model adopted, and thus the mobility model reflecting real-world situations accurately should be obtained [27] and the performance should be also analyzed based on the mobility model. Further study on this kind of mobility model and performance evaluation for various location registration methods will be conducted.

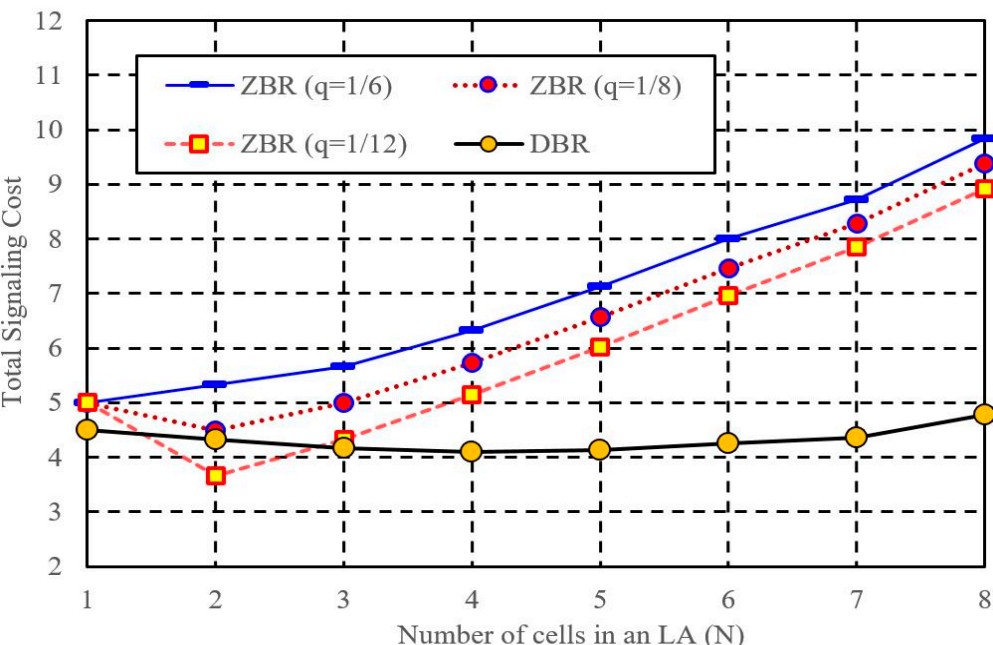

**Figure 17.** Performance comparison results of DBR and ZBR (CMR = 1, U = 4).

## 7. Conclusions and Discussion

This study investigated distance-based registration (DBR), which is known to have a good performance but is not adopted in most mobile communication networks mainly because of difficulty of implementation.

We proposed a DBR with an LA increasing on a cell-by-cell basis, rather than a conventional LA increasing on a ring-by-ring basis, to obtain optimal performance for DBR. We also presented a semi-Markov process model to analyze the optimal number of cells for an LA in DBR with an LA increasing on a cell-by-cell basis. In this study, to construct an LA increasing on cell-by-cell basis, we did not simply limit the center of the LA to the center of the cell. Instead, we tried to have an LA consisting of a number of cells by newly defining the center of the LA to be possible at any point in the LA.

Numerical results for various system environments using the presented semi-Markov process model showed that the optimal *N* with a minimal total signaling cost exists between four and seven in most cases, thus resulting in a smaller total signal cost than that in a ring-by-ring DBR environment (*D* = 2, N = 7) reported in previous studies.

Therefore, when a DBR is adopted, the LA should be constructed to enable an LA to increase on a cell-by-cell basis as shown in this study. Various numerical results showed that the smaller the registration cost compared to the paging cost (i.e., the larger the CMR value), the smaller the size of the LA with a minimal total signaling cost. Results of this study can be used for implementing DBR with excellent performance in real cellular networks.

Of note, this study presents only a conceptual idea of constructing an LA that increases on a cell-by-cell basis (not a ring-by-ring basis) in DBR, and it is not easy to apply the proposed method directly to the design of contemporary multi-RAT (radio access technologies) networks. Further studies will be performed to apply to contemporary multi-RAT networks by taking into account the hierarchical cell layouts, universe frequency bands, complex network procedures and so on.

**Funding:** This research was supported by the Research Base Construction Fund Support Program funded by Jeonbuk National University in 2021. This research was also supported by the Basic Science Research Program through the National Research Foundation of Korea (NRF), funded by the Ministry of Education (2016R1D1A1B01014615).

**Conflicts of Interest:** The authors declare no conflict of interest.

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
