# Peer review of "Improvement and Performance Evaluation When Implementing a Distance-Based Registration"

_applsci, doi:10.3390/app11156823_

Round 1

Reviewer 1 Report

This study proposed a cell-based version of distance-based registration (DBR) (instead of ring based) to improve the total signaling cost.

My comments are the following:

The analysist in this work shall be much more comprehensive than prvious work in [3].

Jung, J., & Baek, J. H. (2020). Analyzing Zone-Based Registration Using a Three Zone System: A Semi-Markov Process Approach. Applied Sciences, 10(16), 5705. doi:10.3390/app10165705 

For example, the explicit comparison in terms of pros and cons of different approaches referenced in section 1 shall be conducted
including the performance analysist of those approaches. 

-In eq 16, the lamda is not defined. 

Reviewer 2 Report

The paper is well structure and the results are very clearly described and illustrated. The problem discussed in the paper is vital for the overall performance of the mobile cellular networks. However, it must be noted, that throghout the years, the architecture of the radio access networks has evolved, and the simplified hexagonal cell models (the starting point for early designs of mobile RANs) is not necesarilly well reflected in real life scenarios. It could be interesting to see also the discussion of the suitability of the proposed approach to the design of contemporary multi-RAT networks, using herarchical cell layouts, diverse frequency bands, MIMO/massive MIMO antenna systems, and complex network procedures. Also, the selection of the signalling load may be additionally affected by the user mobility patterns (e.g. average user speed). It must be noted, the the problem of paging load optimization has been also covered in the research by other teams (e.g. in METIS project). It would be appreciated to see the discussion of the LA-size optimization in the wider RAN planning context - unfortunately, the introduction, the background, and the references are mainly limited to the previous works of the Author.

Round 2

Reviewer 1 Report

As the authors mentioned, "... However, in conclusion, the performance of DBR is always better than that of ZBR in this study, because we adopt a random walk mobility model [9,11,15-16]. If a mobility model favorable to ZBR is assumed as in [6-7], the results may change, and further research on this topic will be conducted." .

To give the complete overiew on the current development of the field to the reader, this topic shall be discussed further in more details. 

I am not sure whether it would be possible to apply real-world data to the analysis on this topic. The reference of this could be found at:

chrome-extension://oemmndcbldboiebfnladdacbdfmadadm/https://www.kdd.org/exploration_files/June_2019_-_1._Urban_Human_Mobility,_Data_Drive_Modeling_and_Prediction_.pdf

Round 3
